# Directed Evolution of 4-Hydroxyphenylpyruvate Biosensors Based on a Dual Selection System

**DOI:** 10.3390/ijms25031533

**Published:** 2024-01-26

**Authors:** Hongxuan Du, Yaoyao Liang, Jianing Li, Xinyao Yuan, Fenglin Tao, Chengjie Dong, Zekai Shen, Guangchao Sui, Pengchao Wang

**Affiliations:** 1School of Life Science, Northeast Forestry University, Harbin 150040, China; dhx171706@163.com (H.D.); liangyaoyao96@163.com (Y.L.); jianingli2002@163.com (J.L.); m472q2@163.com (F.T.); 2NEFU-China iGEM Team, Northeast Forestry University, Harbin 150040, China; ryanryan2023@163.com; 3Key Laboratory for Enzyme and Enzyme-Like Material Engineering of Heilongjiang, College of Life Science, Northeast Forestry University, Harbin 150040, China; 4Aulin College, Northeast Forestry University, Harbin 150040, China; 5School of Pharmacology, China Pharmaceutical University, Nanjing 210009, China

**Keywords:** dual selection system, aromatic compound, directed evolution, PobR, ligand specificity

## Abstract

Biosensors based on allosteric transcription factors have been widely used in synthetic biology. In this study, we utilized the *Acinetobacter* ADP1 transcription factor PobR to develop a biosensor activating the P*pobA* promoter when bound to its natural ligand, 4-hydroxybenzoic acid (4HB). To screen for PobR mutants responsive to 4-hydroxyphenylpyruvate(HPP), we developed a dual selection system in *E. coli*. The positive selection of this system was used to enrich PobR mutants that identified the required ligands. The following negative selection eliminated or weakened PobR mutants that still responded to 4HB. Directed evolution of the PobR library resulted in a variant where PobR^W177R^ was 5.1 times more reactive to 4-hydroxyphenylpyruvate than PobR^WT^. Overall, we developed an efficient dual selection system for directed evolution of biosensors.

## 1. Introduction

Among the numerous valuable aromatic compounds, polyphenols represent a substantial group derived from phenolic substances. In 2020, they reached a market value of $1.6 billion, and their compound annual growth rate (CAGR) is projected to be 5.2% from 2021 to 2030 [1]. Notably, certain compounds with anti-tumor and antibacterial properties, such as resveratrol and ferulic acid, achieved market sizes of 71.9 million and 67.8 million in 2020 and 2022, respectively. They are expected to reach 130 million by 2030 [1]. Additionally, compounds such as tyrosol [2], salidroside [3] and salvianic acid A have gained prominence in the fields of medicine and food health. Phenols are typically and chemically extracted from plants for high product yields. However, these methods may contribute to land overexploitation [4,5]. In recent years, metabolically engineered microbial chassis have emerged as efficient cell factories for the production of phenolic substances. The primary advantage of the use of a microbial cell factory is its ability to generate various chemicals from sustainable raw materials under environmentally favorable conditions [6]. Among various microorganisms, *E. coli* is one of the most widely used hosts for microbial factories due to its rapid growth rate, unique characteristics and well-characterized centralized regulatory system [7]. 4-hydroxyphenylpyruvate (HPP) plays a pivotal role as a fundamental precursor in the biosynthesis of aromatic compounds, and the synthesis of numerous high-value compounds can be achieved based on this premise [1,8] (Figure 1). Monitoring the amount of HPP in chassis cells can provide valuable support for the biosynthesis of these downstream compounds, and the development of biosensors for precursor compounds is beneficial for the synthesis of various downstream compounds [9]. In this regard, the development of biosensors is an important role.

Biosensors based on allosteric transcription factors (aTFs) have emerged as valuable tools for the quantitative detection of small molecule alterations by triggering downstream genetic circuits as outputs [10]. Amounts of intracellularly accumulated compounds are converted into machine-readable outputs such as fluorescence or colorimetric changes, cell density alterations, etc. [11]. Their detectable fluorescence is often used as the output signal in quantitative metabolite determination and high-throughput screening [12,13]. Thus, aTF-based biosensors with innately high ligand specificity are widely used in molecule detection, enzyme-directed evolution, dynamic control of metabolic pathways and adaptive laboratory evolution [14,15,16,17]. Despite their promising applications in synthetic biology, only a handful of TFs have been developed as biosensors due to the limited number of reported ligand compounds and their effector aTFs [18]. Biosensor effectors, such as putrescine [19], naringenin [20], oleic acid [21] and others, have been identified. Nevertheless, biosensors for many aromatic compounds such as HPP, tyrosol and Salvianic acid A are still lacking [11,14,22]. In our previous study, we used PobR as an aTF to develop biosensors. In the ADP1 strain of *Acinetobacter*, when PobR activates the pobA promoter (P*pobA*) involved in 4HB metabolism, low 4HB levels can trigger pobA gene expression [6,14,17]. We employed error-prone PCR to develop a PobR-based biosensor responsive to a 4HB analogue, 4-hydroxymandelate (HMA). Our results showed that PobR possesses the potential to be modified as a biosensor for a variety of aromatic compounds.

As a tool of synthetic biology, transcriptional regulators perform high-throughput screening of large libraries to accelerate the directed evolution and metabolic engineering of enzymes. To screen for strains with improved productivity, several transcriptional regulators have been applied to monitor the production of useful chemicals, such as naringenin [23], diamine [24], isopentanol [25] and butanol [26]. In this regard, it is meaningful to expand the availability of various transcriptional regulators with different ligand specificities. However, desired molecules may not always be recognized as functional ligands by natural transcriptional regulators [27]. Therefore, it is very important to modify their ligand specificities in order to develop and optimize biosensors. For this purpose, the ligand-binding specificity of several allosteric aTFs has been modified through rational mutagenesis or directed evolution to generate customized biosensors for the detection of various compounds of interest. Taking an example, the aTF AraC has evolved its ligand specificity from L-arabinose to D-arabinose, mevalonate and triacetic acid lactone [28,29,30]. An obvious dilemma for the applications of these adapted biosensors is that mutations often alter aTFs’ original ligand docking pockets, leading to relaxed ligand specificity, hindering their use in large-scale screening campaigns. Directed evolution is often employed in the development of novel aTF-based biosensors. The first workflow in directed evolution is to construct a mutant library and then screen it, where the screening usually ought to be high-throughput and fluorescence-activated cell sorting and droplet-sorting screening methods are often employed [31,32,33,34]. However, these methods are equipment-dependent, limiting the development of novel biosensors. In the directed evolution of biosensors, mutants that acquire sensitivity to new ligands tend not to loose sensitivity to their original ligands [35]. In order to obtain a mutant specific to a new ligand, a dual selection system has been applied in previous studies [29,36,37,38]. Using a dual selection system with antibiotic resistance genes and fluorescent protein genes as reporter genes, LuxR variants with required ligand specificity were obtained through directed evolution. Prior to the ligand-responsiveness screening, the mutants responding to the original compound were wiped out by a negative selection [27]. Using this strategy, desirable aTFs can be enriched and identified, and novel biosensors can be developed. In conventional bidirectional screening systems, negative screening typically occurs after the construction of the mutant library, extending the screening process. In the current study, negative screening is completed by exogenous addition of 5-fluorocytosine (5-FC) in the bidirectional screening system, which provides a more efficient and effective screening strategy. In recent years, growth-coupled selection systems had been applied for enzymes [39,40] and strain optimization [41]. They have been proven to be an efficient and convenient way for high-throughput screening of mutant libraries.

In this study, we developed a straightforward, high-throughput, equipment-independent, and versatile dual screening system for identifying biosensors to detect HPP. To accomplish this, we generated a PobR mutant library consisting of 21,000 mutants through a random mutagenesis approach. In our bidirectional screening system, we introduced an innovative negative screening strategy aimed at excluding mutants displaying transcriptional activity in the absence of inducers or in the presence of 4HB. Through this screening process, we successfully identified PobR mutants responsive to HPP and further assessed their responsiveness to various other aromatic compounds. To gain insights into the altered ligand specificity, we made predictions regarding changes in protein structure and ligand binding. In summary, our research introduced an effective and streamlined biosensor selection system within a dual screening framework and pinpointed highly responsive variants of HPP. These findings have implications for the development and enhancement of downstream compounds.

## 2. Results

### 2.1. Design of a Dual Selection System

As an aTF, PobR (or PobR^WT^) drives the promoter P*pobA* and is highly specific for its native effector 4HB. Previous studies indicate that PobR is very difficult to be engineered using any rational approach [34,42,43]. Therefore, we used the random mutagenesis approach to modify its ligand specificity, aiming to reduce its sensitivity to 4HB but increase its responsiveness to other aromatic compounds. To eliminate PobR mutants that were either still responsive to 4HB or constitutively active in driving the P*pobA* independent of any ligand, we designed a dual selection system (Figure 2a). In this system, we first constructed a negative selection system consisting of the promoter P*pobA* and the cytosine deaminase (CDase encoding by *codA*). This was to exclude PobR mutants that retained the ability to respond to 4HB, as well as any false-positive mutant. PobR^WT^ activates the P*pobA* by binding to the operator site (Oi in Figure 2), which activates the expression of downstream *codA* genes [44]. Its product CDase converts exogenously added 5-fluorocytosine (5-FC) to 5-fluorouracil (5-FU), leading to cell death (Figure 2b). Thus, 5-FC sensitivity excluded the false-positive strains that produced PobR mutants capable of activating the *codA* gene driven by the P*pobA* either in the absence of any ligand or through binding to any unknown molecule that activates this promoter. Similarly, bacteria containing PobR mutants that still recognized 4HB perished. The *E. coli* genome contains the *codA* gene, which can produce endogenous CDase. Therefore, we knocked out this gene, using the CRISPR-pCas9 method in the BW25113 strain, and generated a new *E. coli* strain named BW∆*codA*.

In addition, the chloramphenicol (Cm) resistance gene (*cmr*) was placed under the control of the P*pobA.* This was used to select the PobR mutants with the desired ligand specificity. The survival of the Cm-resistant recombinant *E. coli* carrying P*pobA*-codA-cmr and PobR^WT^ depends on the ability of 4HB responsiveness. In the absence of 5-FC in the medium during incubation, the growth of the strain was unaffected regardless of CDase’s expression. Thus, when we added Cm to the LB agar medium containing an aromatic compound and cultivated the PobR mutants from the library, the strains that were responsive to the compound predominantly grew, indicatting a positive selection (Figure 2c).

An additional reporter gene (red fluorescent protein *mCherry)* was also added to the selection system (Figure 2). The reporters were expressed when the PobR protein underwent allosteric changes upon binding to a ligand and then activated the P*pobA* [34]. The red fluorescence intensity of the mCherry protein was proportional to the ligand levels in the medium and could, therefore, be used to determine the binding affinity of the ligand. This dual selection system allowed the enrichment of PobR mutants which were effective in recognizing an aromatic compound as the most preferable ligand.

Based on the above design, we constructed the plasmid gYB2a-pobR^WT^-mCherry-codA-cmr, the PobR^WT^ biosensor, to test the experimental conditions. PobR^WT^ exhibited a high degree of ligand specificity with a narrow dynamic range (0.03–0.50 g/L, Appendix A). Thus, we used 0.5 g/L 4HB to enable the expression of downstream reporter genes. In the presence of 4HB, the growth of *E. coli* BW∆*codA* harboring the PobR^WT^ biosensor was resistant to chloramphenicol. As the concentration of chloramphenicol increased within a certain range, the growth difference between the 4HB-supplemented and non-4HB-supplemented strains gradually widened in bacteria harboring the PobR^WT^ biosensor and supplied with 0.5 g/L 4HB. At high concentrations of chloramphenicol addition, the strain with added 4HB exhibited significant growth advantage, demonstrating the feasibility of a positive screening approach (Figure 2d). Concerning the growth of bacteria in the medium containing either 0.5 g/L 4HB or 5 mg/L 5-FC alone, or both, OD_600_ was measured after 12 h of cultivation. Bacterial growth was inhibited with the addition of 4HB and 5-FC to the medium compared to the other three cases. There was a significant difference in bacterial growth compared to the medium without any addition of 4HB and 5-FC, demonstrating the validity of our negative screening. (Figure 2e). Meanwhile, mCherry expression in the bacteria could be detected by a high level of fluorescence. With the addition of chloramphenicol, the growth of the bacteria was significantly stronger in the presence of 4HB than without it. In the negative-selection simulated experiments, the bacteria growth was repressed in the medium containing 5 mg/L 5-FC and 0.5 g/L 4HB. In contrast, the bacteria generally grew normally in the presence of either 4HB or 5-FC alone. These results verified the effectiveness of the designed dual selection system.

### 2.2. Directed Evolution of PobR

We constructed a random mutagenesis PobR library using the error-prone PCR approach as described in the Methods section. The PobR mutants were subcloned into the vector containing the P*pobA*, *mCherry*, *codA* and *cmr,* using the Golden Gate Assembly system, and then transformed into the BW∆*codA* strain. The storage capacity of the PobR mutagenesis library was determined to be approximately 21,000 clones, with an average mutation rate of about 0.36%.

In the negative selection, the obtained bacteria were cultured in a medium supplemented with 4HB and 5-FC. In the initial negative selection, we used a constant dose of 4HB at 0.5 g/L and then tested different does of 5-FC to inhibit both the false-positive and 4HB-responsive strains. In this step, we first used 50 mg/L 5-FC, and insufficient inhibition of the bacterial was observed. Therefore, we increased the 5-FC concentration to 200 mg/L to increase the selection strength.

In the positive selection, seven valuable aromatic compounds with structural similarity and similar functional groups to 4HB, including HPP, phenylethanol (2-PE), mandelate (MA), 4-hydroxymandelate (HMA), phenylpyruvate (PPA), phenylacetaldehyde (PAld) and p-Coumaric acid, were selected to assess their application as allosteric ligands of PobR mutants. The bacterial cultivation was conducted in the negative selection liquid medium supplied with different aromatic compounds, as well as chloramphenicol. The selection capacity for each compound was more than 900,000 clones (with at least four plates and about 225,000 clones per plate). (Appendix A). We controlled the intensity of selection pressure by adjusting chloramphenicol levels. After negative selection, the PobR mutants were transferred to the LB media containing ampicillin and chloramphenicol, detecting each ligand at a concentration of 0.5 g/L for initial positive selection and single colony isolation. Subsequently, colonies were selected for liquid culture and ligand-responsive PobR mutants were obtained to further characterize the responsiveness to any aromatic compound by evaluating the expression of downstream reporter genes (Figure 3).

### 2.3. Screening for HPP-Responsive PobR Mutants

Following two rounds of negative selection of the mutants, the increased concentration of 5-FC resulted in a significantly reduced response to 4HB of the obtained PobR mutants compared to PobR^WT^. Fluorescence microscopy showed that the mutant strains screened under conditions of 0.5 g/L 4HB and 200 mg/L 5-FC had reduced fluorescence in the presence of 4HB (Figure 4a). Meanwhile, a negative control in the absence of any candidate ligand was used, and almost no false-positive mutant strains were discovered (Figure 5a). In particular, the expression of the mCherry correlated positively with the responsiveness of the tested PobR mutants. Therefore, only pink colonies were selected during the forward screening (Figure 4b), and candidate ligands at 0.5 g/L were used for initial identification.

Several reactive strains were isolated in positive selection experiments using PobR mutants with different ligands. To further assess the obtained PobR mutants, monoclones were isolated through plate stripe treatment. A second round of characterization experiments was conducted to evaluate their reactivity to each candidate ligand. Finally, a PobR variant highly responsive to HPP was identified (Figure 5b,c). Sequencing analysis localized the mutation site to the 177th amino acid, where tryptophan was replaced by arginine. This variant was designated as PobR^W177R^. Simultaneously, a ligand-response curve was generated, showing that the response of PobR^W177R^ to HPP was 5.1-fold higher than that of the control PobR^WT^ (Figure 4c). Compared to the control PobR^WT^, PobR^W177R^ displayed a significant change in specificity and a wider dynamic range of HPP (0.01–1.00 g/L), indicating a reduction in ligand binding affinity.

### 2.4. The Specificity of PobR177 and Screening of PobR Mutants Responsive to Other Ligands

In addition, we evaluated the specificity of PobR^W177R^ in detecting different aromatic compounds. Although 4HB can still be activated and drove mCherry expression, the fluorescence was almost halved compared to the wild type and 4HB, indicating that our bidirectional screening was effective (Figure 4d). However, other aromatic compounds have weak responses to PobR^W177R^. In addition, we found a variant PobR^W177R, L201Q, V225I^, which has a 4.3-fold fluorescence response to HPP (Appendix A), showing significant specific changes with a wide range of changes and low background expression. For other compounds, the results of our forward screening with other ligands showed that another clone (PobR^R40C^) exhibited a rate of maximal induction to PPA (0.5 g/L) over 2 times higher than the ligand-free condition. Importantly, this PobR mutant showed marginal basal transcriptional activity (Appendix A).

### 2.5. Kinetic Simulation and Ligand Docking

In order to better investigate the difference of PobR^W177R^ and PobR^WT^ compared to HPP, homology modelling and docking of the generated mutants were made. Docking of PobR^W177R^ with HPP was stimulated to a very low energy conformation (docking score = −6.3 kcal/mol), which was lower than that of the PobR^WT^ with HPP −5.3 kacl/mol. The speculation is that PobR^W177R^ potentially exhibits a strong binding affinity towards HPP [45]. According to the docking of the results, the PobR^W177R^ protein and HPP formed seven hydrophobic bonds, which is four bonds more than PobR^WT^. It led to increased entropy, decreased energy, and reduced surface tension of the system, which stabilized the whole system (Figure 6a,b). Detailed statistics are available in Appendix A [46].

Kinetic simulation was further used to initially measure the stability of the system. As the RMSD showed (Appendix A), PobR^WT^ binding with 4HB was in a more stable state during the 50 ns simulation, and the overall curve did not fluctuate much. Then, MMGBSA was used to calculate the binding energy [47,48], while the binding energy of PobR^W177R^ with 4HB increased slightly compared to PobR^WT^, from −18.6 kcal/mol to −17.2 kcal/mol. As for HPP, the binding energy increased from −21.64 kcal/mol to −19.98 kcal/mol (Figure 6c,d), which is consistent with our experimental results.

## 3. Discussion

As an intermediate in the synthesis of high-value compounds, HPP currently has no corresponding biosensors. Therefore, the development of biosensors that can respond to HPP is conducive to the synthesis and production of high-value compounds. In the past decades, various approaches have been developed for the directed evolution of aTFs to alter the compound-binding specificity [11]. PobR is an aTF that specifically responds to 4HB [17]. In our previous study, we developed two biosensors responsive to HMA through screening for a library of random PobR mutants [34]. However, in addition to HMA responsiveness, they also responded to their native ligand, 4HB, and other aromatic compounds. It indicates the high potential of PobR variants to become biosensors for various aromatic compounds. Based on these findings, we designed the dual selection system in this study to search for PobR mutants with highly specific responsiveness to different aromatic compounds.

To avoid the loss of the most desirable evolutionary aTFs due to adaptive mutations in host cells, selection-based screening in a solid or liquid medium must be improved to some extent. These cells will gain adaptations that allow them to survive under selective pressures [23]. We developed a dual selection system in *E. coli* using a negative selection to eliminate false-positive strains and those highly responsive to 4HB. In this negative selection, *codA*, a conditional lethal gene, can convert 5-FC to cytotoxic 5-FU with exogenously supplied 5-FC and high 4HB levels. By increasing the 5-FC concentration, we successfully increased the effectiveness of the selection; after two rounds of negative selection, we obtained the mutants lacking both constitutive promoter activity and responsiveness to 4HB.

We used antibiotics in the positive selection of our dual selection system. Our findings indicated that some mutants altered the promoter binding affinity and became constitutively active in driving the promoter. This allowed the bacteria to survive on solid media containing both chloramphenicol and aromatic compounds. Using our dual selection system, we increased the efficiency of a single round of screening to 225,000 clones/plate, which greatly improved the selection efficiency. Compared to other high-throughput selection or screening methods, such as flow cytometry [49], our strategy is equally efficient, but more economic and easier to operate. Using this selection system, we finally obtained the mutant that responded to HPP and PPA the highest responsiveness of 5.12- and 2.01-fold, respectively, which was the proof of concept of our selection system.

In contrast, mutants responsive to HPP and HMA exhibited low ligand specificity in our previous study [34], which could be due to the attenuated allostery and/or distorted ligand binding pocket caused by the multiple-site mutations. Two types of accounts could be proposed for why PobR^W177R^ has better ligand specificity. The first is that ligand specificity alterations from a single mutation are likely more stable. The second is that more hydrophobic bonds were formed between the PobR^W177R^ protein and HPP, which led to increased entropy, decreased energy and decreased surface tension of the system, thereby stabilizing the whole system. To make our system more energetically stable, we optimized the protein structure of the PobR protein model 5HPi on the basis of previous studies [43]. This optimization is promising for the study of the PobR protein in the future. 

In the kinetic simulation, the stability of the system was tentatively measured by the RMSD curve, and it was found that the binding energy of PobR^WT^ and the HPP increased slightly compared with the PobR^W177R^. The increase in binding energy led to a decrease in the stability of the system, which may contribute to a better combination of HPP and PobR^W177R^. Based on our current results, coupled with previous studies, more factors affecting the specificity of transcription factor-based biosensors still need to be investigated and developed.

In summary, we successfully designed and developed a dual selection system for the selection of HPP- and PPA-responsive PobR mutants. The system has great promise for the development of various aromatic biosensors. These biosensors with low detection thresholds and wide dynamic ranges can be of great value for quantitative measurements of valuable aromatic compounds.

## 4. Methods and Materials

### 4.1. Bacterial Strains, Media, Chemicals and Other Materials

The bacterial strains and plasmids used in this study are listed in Appendix A. *E. coli* DB3.1 was used for the construction of the original plasmid gYB2a-pobR-mCherry-codA-cmr. In the negative selection using the *codA* gene, BW∆*codA* was used for the construction and screening of a random mutation library. *E. coli* bacteria were cultured in the Luria–Bertani (LB) medium for propagation. LB liquid medium was prepared by dissolving 10 g tryptone, 5 g yeast extract and 10 g NaCl in 1 L of deionized water, while LB solid medium contained 15 g/L of agar. In the construction of the libraries and mutant screening, *E. coli* bacteria were grown in the M9 medium (17 g Na_2_HPO_4_·7H_2_O, 3 g KH_2_PO_4_, 0.5 g NaCl, 1 g NH_4_Cl, 2 mM MgSO_4_, 0.1 mM CaCl_2_, 4 g/L glucose). The bacteria were cultured in a shaking incubator at 37 °C and 200 rpm. In particular, when cultured in 96-well plates, the culture conditions were 37 °C and 850 rpm. Ampicillin was the conventional antibiotic of choice at a concentration of 100 mg/L. Primers for plasmid construction and PobR random mutagenesis listed in Appendix A were synthesized by Ruibiotech (Harbin, China). All chemicals, including 4HB and other aromatic compounds, were purchased from Aladdin (Shanghai, China).

### 4.2. Construction of the gYB2a-pobR-mCherry-codA-cmr Plasmid

The gYB2a-pobR-mCherry-codA-cmr contains the PobR coding sequence (CDS), an engineered operon consisting of two repetitive P*pobA* promoters, and three CDSs, including the *mCherry* CDS, the cytosine deaminase (*codA*) and the chloramphenicol-resistant gene *cmr*. The PobR CDS and P*pobA* were used in our previous study [34], where the PobR CDS had the codons optimized for *E. coli* preference. In the original construct, a DNA fragment containing the P*pobA* promoters, *mCherry* gene and Sucrose-fructanase encoding gene (*sacB*) was amplified using the plasmid pYP1a-pobR-P*pobA**2-mCherry-sacB from the laboratory stock as a template and the primers *PpobA**2-mc-0311-F and Primer2-0311-R. This fragment has homologous arms to gYB2a at both ends. The linearized vector was obtained by the digestion of gYB2a-ccdB using *Eco*RI and *Kpn*I. The fragment and vector were mixed with the ClonExpress^TM^ II recombinant system (Vazyme, Nanjing, China), followed by transformation into *E. coli* DB3.1 competent cells to obtain the gYB2a-P*pobA**2-mCherry-sacB. The generated gYB2a-P*pobA**2-mCherry-sacB was subsequently digested using *Eco*RI. Amplification of the *cmr* fragment was performed using the pYB1a-eGFP-cmr as a template with the primers Cmr-Gibson-0317-F/R. Both fragments were also assembled using the Gibson Assembly System and resulted in the generation of the gYB2a-P*pobA**2-mCherry-sacB-cmr. In the subsequent experiments, *sacB* performed poorly as a negative selection marker, and thus, it was replaced by the cytosine deaminase (*codA*) gene. The replacement method was performed as follows: pUAM-RE-CD was used as the template and CD-Gibson-0425 F/R was used as primer to amplify *codA*. Using gYb2a-P*pobA**2-mcherry-SacB-cmr as the template, the upstream primer cmr-Gibson-F with a homologous arm downstream of the *codA* fragment and the downstream primer Mc-Gibson-F with a homologous arm upstream of the *oda* fragment were used together to reverse amplify the vector fragment, followed by Gibson Assembly. The Gibson Assembly approach employed in this study was performed using the ClonExpress^TM^ II recombinant system (Vazyme, Nanjing, China). In the final step, gYb2a-P*pobA**2-mCherry-SacB-cmr was used as the vector for *pobR^wt^* as the target fragment to be ligated into the final plasmid by Golden Gate Assembly.

### 4.3. Design and Construction of the PobR Mutant Library

To generate PobR mutants, we developed a library through random mutagenesis of PobR using error-prone PCR amplifications. The primers PobR-P1-BsaI-F and PobR-P2-BsaI-R covering the PobR CDS were used with pLB1s-PobR as a template. The purified PCR products containing various PobR mutants were used as the donor in the following Golden Gate Assembly, while gYB2a-mCherry-codA-cmr was used as the receptor. The donor, receptors and the restriction endonuclease BsaI used to generate the sticky ends were mixed together with BSA, T4 DNA ligase and the corresponding buffer, followed by Golden Gate Assembly. In total, 2.1 μg products were obtained after Golden Gate Assembly. The generated library with highly random PobR mutations was transformed into *E. coli* BW∆*codA* to obtain transformants of mutant plasmids. The PobR mutant library was transformed into BW∆*codA* competent cells and transferred to M9 medium for culture in shaking flasks. A fraction of the products was spread on LB agar plates for colony counting to assess the capacity of the library. Nearly 21,000 transformants were obtained (estimated as 10 ng producing 100 clones). Ten clones were randomly picked to check their PobR CDS regions by DNA sequencing for the quality control.

### 4.4. Counter-Selection Using CDase

The library generated above was inoculated into M9 medium containing 0.5 g/L 4HB and 50 mg/L 5-FC and cultured for 12 h to reach the optical density at 600 nm (OD_600_). Then, the overnight-cultured bacteria were inoculated (1% *v*/*v*) into M9 medium for a second round of negative selection. In this round of selection, the M9 medium was supplemented with 0.5 g/L 4HB and 200 mg/L 5-FC and cultured for 24 h to reach OD_600_ ~0.5. A fraction of the cultured medium was transferred onto LB agar plates to isolate single colonies. Forty-five colonies were picked from the plate and inoculated into LB medium containing ampicillin. After 8–10 h of culturing, 2 μL of bacteria from each well was transferred into 200 μL of M9 medium containing ampicillin and 0.5 g/L 4HB. The strains were cultured in M9 medium without 4HB as negative controls and strains containing PobR^wt^ CDS as positive controls under the same culture conditions. After 12 h of incubation in 96-well plates, OD_600_ and red fluorescence (with an excitation wavelength of 552 nm and an emission wavelength of 600 nm) were measured.

### 4.5. Positive Screening Using Cm

Bacteria after the first step of counter selection were transferred into fresh LB agar plates, each containing 0.5 g/L of different aromatic compounds (HPP, PPA, etc.), with different concentrations (60, 90 and 120 mg/L) of chloramphenicol. The colonies were picked from the plates and inoculated into 600 μL of LB medium with ampicillin in deep-well microplates, followed by culturing in a shaking incubator at 850 rpm and 37 °C for 8–10 h. Then, 2 μL of cultured bacteria from each well was used to inoculate the medium in 96 microplates. Each well had 200 µL of M9 medium containing ampicillin and 0.5 g/L of the corresponding aromatic compounds. As a negative control, the same bacteria were added to the M9 medium without any aromatic compound, while PobR^wt^ grown in culture containing 4HB was used as a positive control. Finally, 200 μL of the cultured bacteria from each well was collected to quantify the OD_600_ and red fluorescence.

### 4.6. Modeling Docking and Kinetic Simulation

For the structure of the PobR protein, we employed Rosetta. We used the original 5hpi structure as a template for homologous modeling predictions of the PobR protein structure. At the same time, the missing loop in the original 5hpi structure which was reported as a candidate PobR protein structure [43] and was repaired using CCD and NGK algorithms, and the energy of the protein itself was subsequently minimized using foldx. We used ORCA 5.0.4 software to calculate the RESP2 charge of the ligand on the basis of the B3LYP generalization and the def2tzvp motif, replacing the am1-bcc charge generated by the antechamber, where the small molecule conformation was chosen as a stable conformation based on the am1-bcc charge to enhance the reliability of the RESP2 charge. Before conducting the formal molecular dynamics simulation, we first performed structure optimization for the small molecular ligand. The basis set used for structure optimization was def2-svp, and the functional used for calculations was b3lyp. Our small molecular system belongs to ordinary organic molecules, so we maintained consistency in the functional used for subsequent single-point energy calculations, which used the def2-tzvp(f) basis set. The obtained results were then input into Multiwfn [50] to obtain more accurate RESP charges. The RESP2 charge was enhanced by calculating the atomic charges suitable for amber/gaff2 force field simulations using the following consensus.
q(RESP2(0.5)) = 0.5q(gas) + 0.5q(solv)

In our simulations, we used the CB-Dock2 (https://cadd.labshare.cn/cb-dock2/index.php, accessed on 22 January 2024) to determine the binding modes of proteins, and the CB-Dock2 search for docking pockets was obtained by tunneling based on the protein surface [45]. In the formal cMD simulation, we mainly used the CHARMM36m force field and employed the PME algorithm to handle the electrostatic interactions between the system components. In the energy minimization step, we initially performed approximately 15,000 minimization steps using the steep algorithm. Subsequently, we applied the CG algorithm for around 2000 additional steps to further minimize the system and bring it to a reasonably stable state. Afterward, we performed a pre-equilibration step of approximately 500 ps in the NVT ensemble. Finally, we conducted a 200 ns classical molecular dynamics simulation. The last 2000 frames of the simulation trajectory were extracted using Gmxmmpbsa to perform MMGBSA-based free energy calculations, yielding reasonable results. To perform the RMSD analysis, we first performed an energy minimization in the amber14sb force field under the condition of am1-bcc charge, and selected a more stable frame as the initial structure to calculate the RESP2 atomic charge and performed a secondary simulation. The water model was a three-point TIP3P, the cutoff was chosen to be 1.0 and the main process of the simulation was divided into the following three stages, energy minimization, restricted kinetic simulation (nvt/npt) and production simulation, in which the hot bath Berendsen was used for the restricted kinetic simulation and the hot bath was used for the 50 ns production simulation of the Parrinello-Rahman algorithm. In the calculation of MMGBSA, the force field ff19SB was chosen for the proteins, while the gaff2 force field was still used for the small molecules, and for the gb calculation part, the igb parameter was chosen as 5 and the saltcon as 0.150. Finally, the molecular binding mode maps were realized using plip [51] and pymol [52]. 

### 4.7. Statistical Analysis

All statistical analyses were performed using the Prism 8.3.0 software (GraphPad, La Jolla, CA, USA). All data are derived from at least three independent experiments. The results are presented as mean with either standard deviation (SD) or standard error of mean (SEM), and sample numbers are indicated unless otherwise noted in the figure legends. Statistical significance calculations comparing two conditions were performed using a two-tailed unpaired Student’s *t*-test. The criterion of statistical significance level is denoted as follows: * *p* < 0.05, ** *p* < 0.01, *** *p* < 0.001.

## 5. Conclusions

In this study, a PobR mutant library was built by means of random mutagenesis. By taking advantages of the *codA* and *cmr* genes, the dual selection system enabled an effective selection to be made from a library containing 21,000 mutants. Through this dual selection system, we evaluated the responsiveness of a series of aromatic compounds to different PobR mutants and obtained multiple candidates for HPP-responsive biosensors. Further studies were performed to identify the mutated amino acids leading to altered ligand binding specificity. The W177R mutation was found to have a strong influence on the ligand binding specificity of PobR by kinetic simulation and ligand docking. Overall, we exemplified a highly efficient dual selection system for aTF-directed evolution and showed the potential of PobR to be engineered as a biosensor for a variety aromatic compounds.

## Figures and Tables

**Figure 1 ijms-25-01533-f001:**
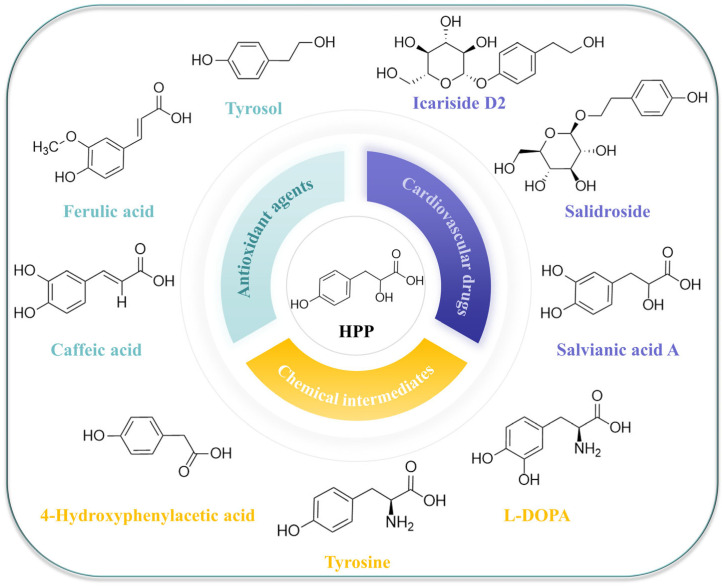
A representative aromatic compound with HPP (4-hydroxyphenylpyruvate) as a precursor in the synthesis pathway and its application diagram. Antioxidant agents: tyrosol, ferulic acid and caffeic acid; Cardiovascular drugs: Icariside D2, Salidroside, Salvianic acid A; Chemical intermediates: Tyrosine, 4-Hydroxyphenylacetic acid, L-DOPA.

**Figure 2 ijms-25-01533-f002:**
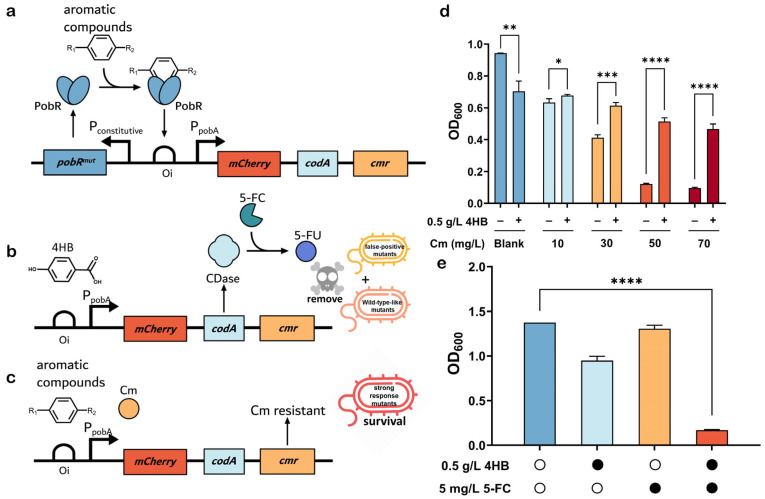
Schematic diagram of the dual selection system and model experiments of negative and positive selection. (**a**) The transcription factor PobR activates the pobA promoter (P_pobA_) when it binds to a ligand. (**b**) When the P_pobA_ is allosterically activated by its native ligand 4HB, CDase is produced and converts the exogenously added nontoxic 5-FC to toxic 5-FU. (**c**) When the P_pobA_ is activated by another aromatic compound, the mutant becomes Cm resistant. (**d**) Growth of bacteria harboring the PobR^WT^ biosensor in medium containing different concentrations of chloramphenicol and supplied with 0.5 g/L 4HB. (**e**) Growth of bacteria in medium containing either 0.5 g/L 4HB or 5 mg/L 5-FC alone, or both. The OD_600_ was measured after 12 h of cultivation. The solid black circle means added and the hollow circle means not added. Each value represents the mean ± standard deviation from 3 biological replicates. *, *p* < 0.05; **, *p* < 0.01; ***, *p* < 0.001; ****, *p* < 0.0001.

**Figure 3 ijms-25-01533-f003:**
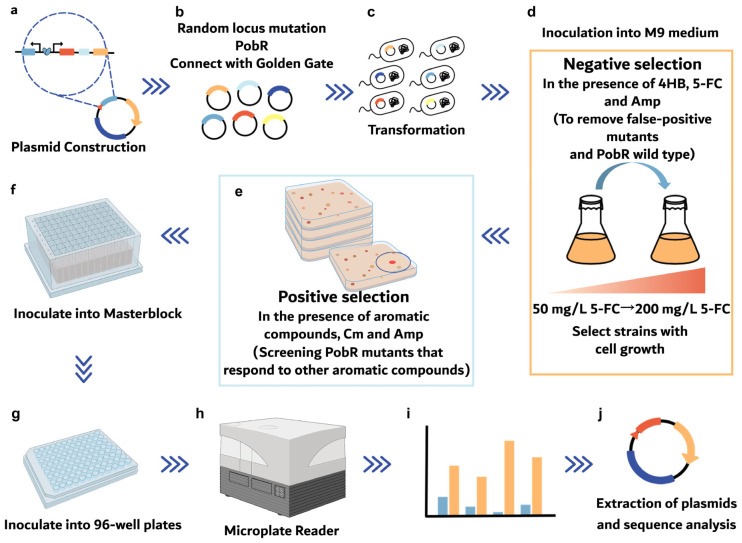
Schematic diagram of the selection procedures for ligand-responsive PobR mutants by eliminating false-positive clones, functional characterization and sequence identification. (**a**–**c**) Construction of a random PobR mutagenesis library generated by the Golden Gate Assembly system and transformed into the *E. coli* BW∆*codA*. (**d**) Negative selection of bacteria in the liquid medium supplemented with 4HB and 5-FC. (**e**) Positive selection of the bacteria was obtained by step b in liquid medium containing different aromatic compounds, including 2-PE, MA, HMA, PAld, HPP and PPA, as well as Cm. (**f**,**g**) Inoculation of selected bacteria into Masterblock followed by transfer into 96-well plates. (**h**–**j**) Determination of fluorescence intensity and altered sequences of the PobR mutants.

**Figure 4 ijms-25-01533-f004:**
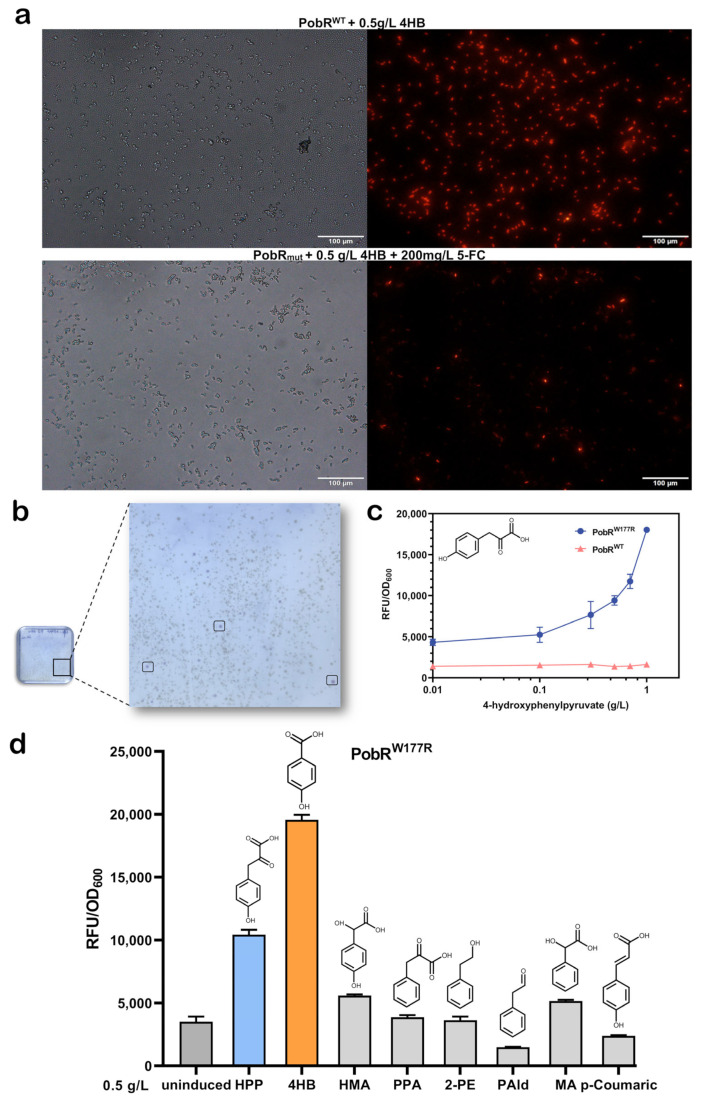
The optimal mutant PobR^W177R^ was obtained after the dual screening. (**a**) The PobR^WT^ *E*. *coli* under fluorescence microscope expressed strong fluorescence under 0.5 g/L 4HB, while the majority of the PobR-mutant *E. coli* showed no/weak fluorescence under the fluorescence microscope with 0.5 g/L 4HB and 200 mg/L 5-FC. (**b**) After two rounds of rescreening, the mutant was coated on a plate containing HPP and chloramphenicol, and some pink mono-clones were grown on the plate 12–24 h post culture. (**c**) Comparison of the ligand-response curves between PobR^W177R^ and PobR^WT^. The vertical axis shows the ratio of *mCherry* (RFU) expression to *E. coli* growth (OD 600) measured 12 h post culture in M9 medium with different doses of HPP. (**d**) Fluorescence changes in PobR^W177R^ in response treatments with different aromatic compounds (similar structure to HPP). The horizontal axis represents the ratio of *mCherry* (RFU) expression to *E. coli* growth (OD 600) measured 12 h post culture in M9 medium containing different ligands.

**Figure 5 ijms-25-01533-f005:**
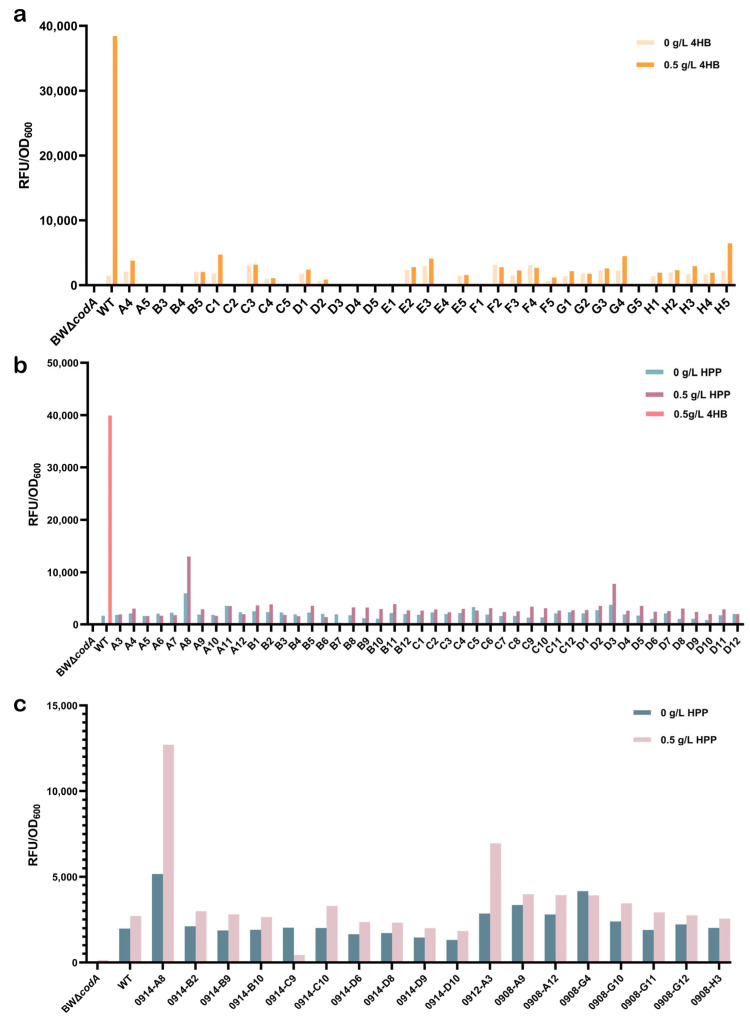
Fluorescence characterization of mutants. (**a**) Response of the PobR strains to 4HB after two rounds of negative selection (50 mg/L and 200 mg/L). Thirty-five single colonies were randomly picked after diluted bacterial culture was spread on LB plates containing only ampicillin. They were activated by LB medium for 10 h and then transferred to M9 medium containing 0.5 g/L 4HB for 12 h of cultivation. (**b**) Evaluation of individual clones for their responsiveness to HPP. The pink monoclonal was selected, inoculated into LB from the forward screening plate, and cultured in M9 containing 0.5 g/L HPP for 12 h for identification. (**c**) After preliminary screening and characterization, the colonies were collected, striated and appeared to be monoclonal, and then HPP was added for characterization. The results showed that 0914-A8(PobR^W177R^) had higher difference in fluorescence characterization. The colonies or clones are denoted by their screening/selection identification numbers. The mean fold induction in specific *mCherry* fluorescence in response to the presence of ligand serves as a measure to compare the PobR mutant biosensors. The concentration of the compounds was 0.5 g/L. BWΔ*codA* was the background control and WT was the positive control.

**Figure 6 ijms-25-01533-f006:**
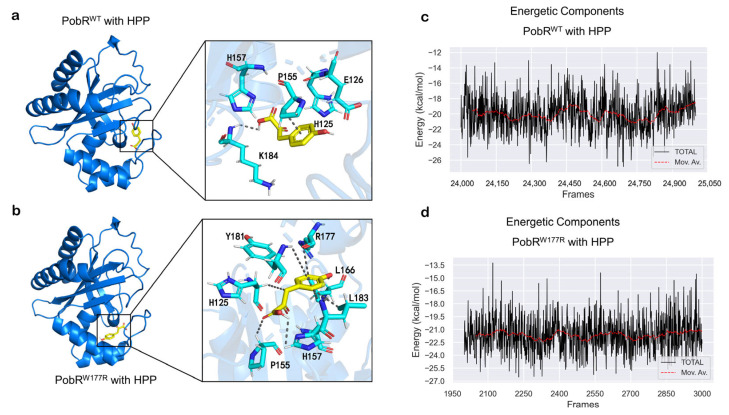
The molecular docking and molecular dynamics simulation of the optimal mutant PobR^W177R^. (**a**) Ligand binding site of PobR^WT^ and HPP is illustrated. In the PobR^WT^ complex, HPP forms salt bridges with His125 and His157 and hydrogen bonds with Glu126 and Lys184. The gray dashed line denotes the hydrophobic force. Amino acid numbering is based on the native PobR sequence, and the yellow compound represents HPP. (**b**) The figure shows the binding site of Pob^W177R^ and HPP. In the PobR^W177R^ complex, HPP forms salt bridges with His157 and hydrogen bonds with His157, as well as hydrophobic interactions with His125, Pro155, His157, Leu166, Arg177, Tyr181, Leu183. The gray dashed line represents hydrophobicity, and the mutation site 177 amino acids participate in the formation of hydrophobic bonds. Amino acid numbering is based on native PobR sequence, and the yellow compound indicates HPP. (**c**) Gromacs and gmx_MMPBSA were used to calculate the binding energy of MMGBSA PobR^WT^ to HPP. (**d**) The method of calculating the binding energy of PobR^W177R^ to HPP was the same as before. Since the calculation of binding energy requires relatively smooth trajectories, the smoothing time of the trajectory of the wild type and mutant protein is different. It is understandable that the time range chosen for the calculation of the binding energy is different.

## Data Availability

Data is contained within the article and Appendix A.

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
