# Peer review of "Directed Evolution of 4-Hydroxyphenylpyruvate Biosensors Based on a Dual Selection System"

_ijms, 2024, doi:10.3390/ijms25031533_

Round 1

Reviewer 1 Report

Comments and Suggestions for Authors

In this article, the authors develop new HPP-responsive biosensors using directed evolution. The protocol included negative selection -- the original ligand or absence of ligand launched expression of gene that synthesized cytosine deaminase that converted exogenously added 5-fluorocytosine to 5-fluorouracil leading to cell death. On the other hand, expression with desired ligand launched chloramphenicol resistant gene, and strains responsive to the compound could grow predominantly. Positive and negative selection allowed to enhance the reactivity to 4-hydroxyphenylpyruvate and eliminate the reactivity to 4-hydroxybenzoic acid of the wild-type protein. The suggested protocol was highly effective and can be utilized for the development of a large number of biosensors reactive to different compounds. The manuscript can be accepted for publication.

Minor

- Line 101: lose -> loose.

- The data presented in Figures 2d, 2e should be described in more detail.

- References in upper indexes should be switched with dots and commas

- In methods: authors say that protein structure was predicted with homology modeling. More information on model preparation should be added -- what was the template; how the model was treated before calculation of RESP charges (MM, QM/MM optimization?), more details on MD simulation.

Author Response

Response to Reviewer 1 Comments

Thanks very much for taking your time to review this manuscript. Please find my itemized responses in below, the reviewer comments are laid out below in italicized font and specific concerns have been numbered. Our response is given in normal font and changes/additions to the manuscript are given in highlighted text.

Comments and Suggestions for Authors

In this article, the authors develop new HPP-responsive biosensors using directed evolution. The protocol included negative selection -- the original ligand or absence of ligand launched expression of gene that synthesized cytosine deaminase that converted exogenously added 5-fluorocytosine to 5-fluorouracil leading to cell death. On the other hand, expression with desired ligand launched chloramphenicol resistant gene, and strains responsive to the compound could grow predominantly. Positive and negative selection allowed to enhance the reactivity to 4-hydroxyphenylpyruvate and eliminate the reactivity to 4-hydroxybenzoic acid of the wild-type protein. The suggested protocol was highly effective and can be utilized for the development of a large number of biosensors reactive to different compounds. The manuscript can be accepted for publication.

 Response: Thanks for your positive comments. We are pleased to hear that you found our research meaningful. We would like to thank the referee once more for sparing the time to write so many detailed and useful comments.

Minor concerns:

Q1. Line 101: lose -> loose.

A1. We sincerely thank the reviewer for careful reading, thank you for your great efforts in this detail. As suggested by the reviewer, we have corrected the “lose” into “loose”. (Line 101, page 3)

 Q2. The data presented in Figures 2d, 2e should be described in more detail.

A2. We think this is an excellent suggestion. Thank the reviewer again. We have described the data presented in Figure 2d,2e in more detail, the exact location where the change can be found in the revised manuscript (Line 183-193, page 5).

Q3. References in upper indexes should be switched with dots and commas.

A3. Thank you for very detailed suggestion. The presentation of references throughout the article has been modified in accordance with the comments, the references in the upper index have been switched with dots and commas.

Q4. In methods: authors say that protein structure was predicted with homology modeling. More information on model preparation should be added -- what was the template; how the model was treated before calculation of RESP charges (MM, QM/MM optimization?), more details on MD simulation.

A4. Thank you for your suggestions. We used the original 5hpi structure as a template for homologous modeling predictions of PobR protein structures (Line 486-487, page13),We have added details of the treatment of the model before calculating RESP charges. (Line 494-500, page14) Here, we have added literature to support these details (Reference52: Lu, T.; Chen, F. J. J. o. C. C. O., Inorganic, Physical, Biological, Multiwfn: A multifunctional wavefunction analyzer. 2012, (5), 33). At the same time, we have added more details on MD simulation in the Methods section to the manuscript (Line 506-515, page14).

Once again, thank you for taking the time and effort to review my article. I am grateful for your support, and I deeply appreciate your dedication to the academic community.

Reviewer 2 Report

Comments and Suggestions for Authors

The manuscript Directed Evolution of 4-hydroxyphenylpyruvate Biosensors Based on a Dual Selection Systemdescribes the development of a simple, high-throughput, equipment-independent, and versatile dual screening system for the identification of biosensors for the detection of 4-hydroxyphenylpyruvate (HPP). HPP plays a key role as a fundamental precursor in the biosynthesis of aromatic compounds, based on this premise, the synthesis of numerous high-value compounds can be achieved. Monitoring the amount of HPP in cells can provide valuable support to the biosynthesis of these downstream compounds, and the development of biosensors for precursor compounds is useful for the synthesis of various valuable compounds in downstream steps. The authors developed a dual screening system to search for PobR mutants with highly specific responses to various aromatic compounds. Using this system, the authors obtained a PobR mutant that responded to HPP and PPA with the highest efficiency of 5.12- and 2.01-fold, respectively, providing proof of concept for the selection system.

The manuscript clear, relevant for the field and presented in a well-structured manner. The cited references contain enough of recent publications (within the last 5 years) and cited references are relevant. There is not excessive number of self-citations. The manuscript scientifically sound and the experimental design appropriate to test the hypothesis. The results reproducible based on the details given in the methods section.

The figures and schemes are appropriate, properly show the data, and are easy to interpret and understand. The data interpreted appropriately and consistently throughout the manuscript. The conclusions consistent with the evidence and arguments presented.

Author Response

Response to Reviewer 2 Comments

Comments and Suggestions for Authors

The manuscript “Directed Evolution of 4-hydroxyphenylpyruvate Biosensors Based on a Dual Selection System” describes the development of a simple, high-throughput, equipment-independent, and versatile dual screening system for the identification of biosensors for the detection of 4-hydroxyphenylpyruvate (HPP). HPP plays a key role as a fundamental precursor in the biosynthesis of aromatic compounds, based on this premise, the synthesis of numerous high-value compounds can be achieved. Monitoring the amount of HPP in cells can provide valuable support to the biosynthesis of these downstream compounds, and the development of biosensors for precursor compounds is useful for the synthesis of various valuable compounds in downstream steps. The authors developed a dual screening system to search for PobR mutants with highly specific responses to various aromatic compounds. Using this system, the authors obtained a PobR mutant that responded to HPP and PPA with the highest efficiency of 5.12- and 2.01-fold, respectively, providing proof of concept for the selection system.

The manuscript clear, relevant for the field and presented in a well-structured manner. The cited references contain enough of recent publications (within the last 5 years) and cited references are relevant. There is not excessive number of self-citations. The manuscript scientifically sound and the experimental design appropriate to test the hypothesis. The results reproducible based on the details given in the methods section.

The figures and schemes are appropriate, properly show the data, and are easy to interpret and understand. The data interpreted appropriately and consistently throughout the manuscript. The conclusions consistent with the evidence and arguments presented.

Response: Thanks for your positive comments. We are pleased to hear that you found our research meaningful and your acknowledgment of the significance of our work is greatly appreciated. I am grateful for your support, and I deeply appreciate your dedication to the academic community.

Reviewer 3 Report

Comments and Suggestions for Authors

1. Line 40, "Salidroside" and "Salvianic acid A", do not require capitalizing the first letters.

2. Figure 1 contains errors in chemical formulas, including HPP and Salvianic acid A. Additionally, the difference between 4-Hydroxyphenylpyruvic acid and 4-Hydroxyphenylpyruvate should be noted.

3. Line 74 (also other lines in the manuscript), the "Acinetobacter strain" is the wrong way to describe a stain, as Acinetobacter is the name of a Genus. In contrast, the stain usually refers to a Species.

4. Line 141, what does the "Oi" in the "cis-acting element Oi" mean?

5. Lines 147-150, are there any concerns with knocking out the native codA gene in the E. coli genome, in terms of cell growth or other metabolic activity?

6. Figure 2, please label the PobR in Figure 2a; please make it clear what's the effects when adding 4HB or aromatic compounds in Figure 2b and 2c; what does the Cm in Figure 2c do; does the solid circle mean yes in Figure 2e, could you please make it easier to understand?

7. Lines 195-197, how was the storage capacity estimated?

8. It appears that the zoomed-in insets in Figures 6a and b are not aligned with the original protein structure. Could this be due to a difference in orientation? Can you please help me understand this better?

Comments on the Quality of English Language

1. Non-native English words such as "pseudo-positive mutant" (line 140) should be changed to "false-positive mutant".

2. Suggestion: Have a native language speaker check the grammar and writing. Some long sentences are difficult to understand.

Round 2

Reviewer 3 Report

Comments and Suggestions for Authors

The authors have nicely addressed my concerns and suggestions.

Comments on the Quality of English Language

The Quality of the English Language has been improved.